# Acidifiers Attenuate Diquat-Induced Oxidative Stress and Inflammatory Responses by Regulating NF-κB/MAPK/COX-2 Pathways in IPEC-J2 Cells

**DOI:** 10.3390/antiox11102002

**Published:** 2022-10-10

**Authors:** Qinglei Xu, Mingzheng Liu, Xiaohuan Chao, Chunlei Zhang, Huan Yang, Jiahao Chen, Chengxin Zhao, Bo Zhou

**Affiliations:** 1College of Animal Science and Technology, Nanjing Agricultural University, Nanjing 210095, China; 2Yantai Jinhai Pharmaceutical Co., Ltd., Yantai 265323, China

**Keywords:** acidifier, oxidative stress, reactive oxygen species, IPEC-J2, NF-κB/MAPK/COX-2 signaling pathway

## Abstract

In this study, we evaluated the protective effects and potential mechanisms of acidifiers on intestinal epithelial cells exposure to oxidative stress (OS). IPEC-J2 cells were first pretreated with 5 × 10^−5^ acidifiers for 4 h before being exposed to the optimal dose of diquat to induce oxidative stress. The results showed that acidifiers attenuated diquat-induced oxidative stress, which manifests as the improvement of antioxidant capacity and the reduction in reactive oxygen species (ROS) accumulation. The acidifier treatment decreased cell permeability and enhanced intestinal epithelial barrier function through enhancing the expression of claudin-1 and occludin in diquat-induced cells. Moreover, acidifier treatment attenuated diquat-induced inflammatory responses, which was confirmed by the decreased secretion and gene expression of pro-inflammatory (TNF-α, IL-8) and upregulated anti-inflammatory factors (IL-10). In addition, acidifiers significantly reduced the diquat-induced gene and protein expression levels of COX-2, NF-κB, I-κB-β, ERK1/2, and JNK2, while they increased I-κB-α expression in IPEC-J2 cells. Furthermore, we discovered that acidifiers promoted epithelial cell proliferation (increased expression of *PCNA* and *CCND1*) and inhibited apoptosis (decreased expression of *BAX*, increased expression of *BCL-2*). Taken together, these results suggest that acidifiers are potent antioxidants that attenuate diquat-induced inflammation, apoptosis, and maintain cellular barrier integrity by regulating the NF-κB/MAPK/COX-2 signaling pathways.

## 1. Introduction

At weaning, piglets are affected by many stress factors, such as the change of feed and environment, mixing group and so on, which lead to the decline of body immunity and weaning stress syndrome [1,2]. In the process of early weaning, piglets’ antioxidant capacity decreases and free radicals increase, forming oxidative stress in the body, which is considered to be an important reason for “early weaning syndrome”. The intestinal epithelium comprises monolayer cells and is the primary site of oxidative stress response [3]. At the same time, it also has the functions of nutrient absorption and innate immunity, which are very important for maintaining intestinal homeostasis [4]. The tight junction between intestinal epithelial cells is one of the important structural bases for maintaining the function of the intestinal barrier [5]. Recent studies have shown that oxidative stress destroys the tight junction between intestinal epithelial cells in a variety of ways, resulting in intestinal epithelial barrier dysfunction [6,7]. Some oxidative stressors increase the permeability of the intestinal epithelial barrier by disrupting tight junctions, accompanied by the disturbance of the NF-kappaB pathway and intestinal inflammation [8]. Therefore, treatment to maintain the integrity of the intestinal epithelial barrier may be beneficial to intestinal and host health to reduce inflammatory response and oxidative stress injury.

Acidifiers are natural organic acids and salts that have long been considered as the feed additives to replace antibiotics in weaned pigs [9]. Acidifiers improved growth performance [10], alleviated gut inflammation, and enforced gut barrier integrity in pigs [11]. It was reported that acidifiers had a higher antioxidant capacity under heat stress conditions in pigs [12]. Furthermore, studies have demonstrated that acidifiers exert their immune regulatory function and inhibit the invasion of enterotoxigenic Escherichia coli F4 in vitro [13,14]. Our previous study also showed that supplementing the drinking water with acidifiers had the potential as antioxidants, which was reflected in the improvement of growth performance, immunity, and antioxidant capacity in pigs [15]. However, the precise molecular mechanism of acidifier action still remains unclear. On the other hand, most studies have focused on its effects in vivo, and little is known about the underlying protective mechanisms of acidifiers in porcine small intestinal cells, which are related to reducing inflammatory responses as well as improving antioxidant capacity. IPEC-J2 cells were isolated from the middle jejunum of neonatal piglets [16]. In recent years, IPEC-J2 cells have been widely used in studying the intestinal functions associated with immune response [17], barrier function [18], and antioxidant capacity research in vitro [19].

Diquat (DQ) is a commonly used oxidative stress inducer, which produces reactive oxygen species (ROS) and reactive nitrogen species (RNS) through redox cycling processes, and then leads to oxidative stress and cell death [20,21]. Therefore, to clarify the mechanism of the antioxidant effect of the acidifiers, we explored the protective effect of acidifiers on intestinal porcine epithelial cells (IPEC-J2) that were exposed to oxidative stress induced by DQ to prove its effectiveness even when administered in vitro. We also investigated the potential important role of acidifiers in maintaining the integrity of the intestinal epithelial barrier and alleviating the inflammatory responses induced by DQ in the IPEC-J2 cells. Here, we report convincing evidence that acidifiers have antioxidant effects and immune function and provide key insights into their potential mechanisms of action.

## 2. Materials and Methods

### 2.1. IPEC-J2 Cells Culture

The intestinal porcine enterocyte cell (IPEC-J2) lines were kindly provided by the lab of Dr Chunmei Li (Nanjing Agricultural University). The IPEC-J2 cells were cultured and maintained in Dulbecco’s modified Eagle’s medium/nutrient mixture F12 (DMEM/F12) medium (Gibco, Grand Island, NY, USA) supplemented with 10% fetal bovine serum (FBS, Gibco, Grand Island, NY, USA) and penicillin-streptomycin (Gibco, Grand Island, NY, USA). The resuscitated IPEC-J2 cells were cultured in a humidified incubator at 37 °C with a 5% CO_2_ atmosphere. After 12 h, the culture medium was replaced for the first time with fresh medium. After that, the medium was replaced every 24 h depending on the developmental state of the cells. Then, when they reached 80% confluence, the cells were detached with 0.05% trypsin (Gibco, Grand Island, NY, USA) and subcultured in culture medium.

### 2.2. Establishment of Oxidative Stress Model in IPEC-J2 Cells

The DQ-induced oxidative stress model was used to evaluate cell proliferation and cytotoxicity utilizing an MTT cell assay kit (Jiancheng Bioengineering Institute, Nanjing, China). The IPEC-J2 cells grown at a logarithmic phase were treated with 0.1% trypsin to prepare a single cell suspension and then seeded in a 96-well cell culture plate. The number of seeds was 1 × 10^4^ cells/well. The cells were cultivated in an incubator at 37 °C with a constant concentration of 5% CO_2_ for 24 h. The medium was then discarded, and the wells were washed with sterile phosphate-buffered saline (PBS, Gibco, Grand Island, NY, USA) before being filled with medium containing 0, 100, 250, 500, 750, 1000, 1250, and 1500 μmol/L of DQ. Six hours was selected as the treatment time of diquat to induce the cytotoxicity of IPEC-J2 cells according to a previous report [22]. Then, 50 µL of MTT assay solution was added into each well, and it was incubated for 4 h. Afterward, a microplate reader (Tecan, Austria GmbH, Grödig, Austria) was used to determine the absorbance of the plate at 570 nm. In order to establish the cellular oxidative stress model for IPEC-J2 cells, the 50% inhibitory concentration (IC50) was determined as the optimal DQ concentration, causing a 50% reduction in cell viability based on the MTT assay. MTT assays were performed with eight replicate wells per dose and repeated twice to confirm the results.

### 2.3. Treatment of IPEC-J2 Cells with Acidifiers

The optimal concentration and time of acidifiers were determined using a cell counting kit (CCK-8 kit, Jiancheng Bioengineering Institute, Nanjing, China) according to the instruction of the manufacturer. The water-soluble liquid acidifiers (Jinlisuan, Yantai Jinhai Pharmaceutical Co., Ltd., Yantai, China) consist of 19% formic acid, 19% acetic acid, 15% lactic acid, 3.5% propionic acid, and its organic acid salts. The IPEC-J2 cells were treated with the medium containing 2 × 10^−3^, 1.5 × 10^−3^, 1 × 10^−3^, 5 × 10^−4^, 1 × 10^−4^, or 5 × 10^−5^ dilutions of acidifiers using the same seeding method as MTT for 2 h, 4 h, or 6 h. A total of 10 µL of CCK-8 solution was added to each well, and the plate was incubated for 1 h. The optimal acidifier concentration was calculated according to the cell viability based on the CCK-8 assay. The microplate reader (Tecan, Austria GmbH, Grödig, Austria) was used to determine the absorbance of the plate at 450 nm.

To estimate the antioxidant effect of acidifiers on IPEC-J2 cells, the cells were further allocated one of four groups: CON (without any treatments); DQ (DQ treatment, treated with optimal concentration and incubation time based on the above experiment); AC (acidifier treatment, treated with optimal concentration based on the above experiment); and AC + DQ (acidifier pretreatment and then DQ treatment) groups. Before the acidifier or DQ treatment, each group was washed twice with PBS at the same time.

### 2.4. Intracellular ROS Assays

A ROS assay kit (Jiancheng Bioengineering Institute, Nanjing, China) was used to detect the intracellular ROS level of the treated IPEC-J2 cells. The 2′, 7′-dichlorohydro-fluorescein diacetate (DCFH-DA) is the most sensitive and commonly used probe for detecting intracellular ROS. The IPEC-J2 cells were cultured in 96-well plates (10^4^ cells per well, 8 replicates per group) and subjected to their indicated treatments. Next, the cells were washed twice with PBS and treated with 100 μL of 10 μM DCFH-DA (1:1000 dilution in serum-free DMEM/F12 medium) at 37 °C for 20 min. DCFH is oxidized into a strong green fluorescence substance, dichlorofluorescein (DCF), in the presence of ROS in cells, and its optimal excitation wavelength is 488 nm, while the emission wavelength is at 525 nm. Its fluorescence intensity is proportional to the ROS levels in cells. The fluorescence signals were monitored using a microplate reader (Tecan, Austria GmbH, Grödig, Austria).

### 2.5. Antioxidant Indicators and Cytokine Assay

After the cells were treated as described above, they were gently washed twice with PBS and lysed using a Radio Immunoprecipitation Assay lysis buffer (RIPA, Biosharp, Beijing, China) containing 1% phenylmethylsulfonyl fluride (PMSF, Biosharp, Beijing, China) on ice for 20 min. The cells were centrifuged at 10,000× *g* for 10 min at 4 °C, and the cell lysates were gathered to determine oxidative stress indicators: methane dicarboxylic aldehyde (MDA), total anti-oxidation capacity (T-AOC), total superoxide dismutase (T-SOD), catalase (CAT), and glutathione peroxidase (GSH-Px). Cell culture supernatant was harvested and determined the secretion of cytokine, including tumor necrosis factor-α (TNF-α), interleukin-8 (IL-8), and interleukin-10 (IL-10). All antioxidant indicators and inflammatory factors were determined using enzyme-linked immunosorbent assay (ELISA) kits (Jiancheng Bioengineering Institute, Nanjing, China) according to the manufacturer’s guidelines. In brief, we measured the absorbance (OD value) of each well and calculated the test samples according to the standard curve. For antioxidant indicators, the OD values of T-AOC, T-SOD, GSH-Px, MDA, and CAT were measured using a microplate reader (Tecan, Austria GmbH, Grödig, Austria) at 593 nm, 550 nm, 405 nm, 532 nm, and 405 nm, respectively. For cytokine, the OD values of TNF-α, IL-8, and IL-10 were measured at 450 nm. Each group had 6 replicates, and each sample was determined three times. The intra- and inter-assay coefficients of variation were less than 10%.

### 2.6. Immunofluorescence Analysis

The distribution of the tight-junction protein (claudin-1) in the IPEC-J2 cells was determined by an immunofluorescence analysis. Briefly, the IPEC–J2 cells were seeded on cover-slides treated with poly(L-lysine) (Biosharp, Beijing, China) and placed in 12-well plates for 12 h to reach 70% confluence. The cells were fixed with 4% paraformaldehyde (Beyotime, Shanghai, China) for 30 min, and then they were permeabilized with 0.5% Triton X-100 buffer (Beyotime, Shanghai, China) at room temperature for 20 min. Thereafter, the IPEC–J2 cells were incubated with anti-claudin-1 antibody (dilution 1:500; ABclonal, Wuhan, China) for 1 h at room temperature and then incubated with fluorescein conjugated goat anti-rabbit IgG (H + L) antibody (dilution 1:500; Proteintech, Wuhan, China) in the dark for 1 h. The cell nuclei were stained with 4′,6-Diamidino-2-Phenylindole (DAPI, Beyotime, Shanghai, China) solution. The slides were visualized under a laser scanning confocal microscope (Zeiss, LSM 700; Oberkochen, Germany).

### 2.7. Cell Proliferation Assay

IPEC-J2 cells proliferation was quantified using an ethynyldeoxyuridine (EdU) kit (RiboBio, Guangzhou, China) according to the manufacturer’s instruction. The cells were seeded into 12-well plates, as described above. Each well received 300 μL of 50 μM EdU, and the cells were incubated for an additional 2 h, after which they were washed with 500 µL of PBS and fixed with 4% paraformaldehyde for 30 min. To neutralize the excess aldehyde groups, 200 µL of 2 mg/mL of glycine was aliquoted per well and incubated with the cells for 5 min. Subsequently, 500 µL of PBS (0.5% Triton X-100) was added into each well to incubate with the cells for 10 min. After the cells were washed with PBS, 300 μL of Apollo reagent was added, and the cells were incubated in the dark for 30 min at room temperature. The cells were washed with PBS (0.5% Triton X-100) and the nuclei were stained with Hoechst 33342 reaction solution for 30 min in the dark. The EdU-stained cells were visualized and quantified using a fluorescence microscope. Three fields were randomly selected for quantification and statistical analysis.

### 2.8. Detection of Apoptosis by Flow Cytometry

Cellular apoptosis was assessed using Annexin V-fluorescein isothiocyanate (FITC) with a propidium iodide (PI) staining assay (Vazyme, Nanjing, China) according to the manufacturer’s instructions. Briefly, the cells were harvested and incubated with 5 μL of AnnexinV-FITC and PI for 10 min at room temperature in the dark. After that, the cells were resuspended in 400 µL 1 × binding buffer and then mixed thoroughly. Subsequently, the apoptotic cells were immediately measured by a BD FACS Calibur flow cytometer (BD Biosciences, San Jose, CA, USA) and analyzed with FlowJo software (Tree Star, Stanford University, CA, USA).

### 2.9. RNA Extractions and Real-Time Quantitative PCR (qRT-PCR)

Total RNA was extracted utilizing TRIzol Reagent (Invitrogen, Carlsbad, CA, USA) from IPEC-J2 cells. The Nanodrop spectrophotometer (Thermo Scientific, Waltham, MA, USA) was used to examine the RNA concentration and quality. We utilized the HiScript ^®^ III RT SuperMix for qPCR (+gDNA wiper) (Vazyme, Nanjing, China) to synthesize cDNA according to the manufacturer’s instructions. The qRT-PCR was conducted using ChamQ Universal SYBR qPCR Master Mix (Vazyme, Nanjing, China) based on the manufacturer’s manual. All primers were synthesized commercially by TsingKe Biotech Co., Ltd. (Tsingke, Nanjing, China) and are shown in Appendix A. Glyceraldehyde-3-phosphate dehydrogenase (GAPDH) was used as an endogenous control. The relative abundance of each mRNA was calculated using the 2^−∆∆Ct^ method [23].

### 2.10. Western Blotting

Proteins were extracted from the IPEC-J2 cells using RIPA lysis buffer (Biosharp, Beijing, China). Protein concentrations were determined by a BCA protein quantification kit (Vazyme, Nanjing, China) and a microplate reader (Tecan, Austria GmbH, Grödig, Austria). Equal amounts of proteins were resolved on 4–20% sodium dodecyl sulfate polyacrylamide gel electrophoresis (SDS–PAGE, Genscript, Nanjing, China), and then transferred onto a polyvinylidene fluoride (PVDF) membrane (Immobilon Transfer Membranes, Merck Millipore, Merck KGaA, Darmstadt, Germany) with a wet transfer system (Tanon, Shanghai, China). After being blocked with TBST containing 5% non-fat milk powder (Mulinsen, Nanjing, China) for 1 h at room temperature, the membranes were incubated in specific primary antibodies, COX-2, NF-κB, I-κB-α, I-κB-β, ERK1/2, and JNK2 MAPK (1:500, ABclonal, Wuhan, China) at 4 °C overnight. The PVDF membranes were washed with TBST and then incubated with secondary antibody anti-rabbit (1:8000; Affinity, Changzhou, China) for 1 h at room temperature. Signal densities of the immunoblotting image were determined using a high-sensitivity chemiluminescence ECL detection kit (Vazyme, Nanjing, China) on a ChemiDoc^TM^ Imaging System (Bio-Rad, Hercules, CA, USA). Protein levels were normalized to GAPDH, and the densitometric quantification of Western blotting bands was analyzed using ImageJ software (NIH, Bethesda, MD, USA).

### 2.11. Statistical Analysis

The data were analyzed using a one-way analysis of variance (ANOVA) and Tukey’s test to determine the differences between treatments in SAS 9.4 software (SAS Inst. Inc., Cary, NC, USA). The results are expressed as means ± standard error of the mean (SEM). The calculation of median effective dose (IC50) was conducted using GraphPad Prism software (GraphPad Prism v8.0, GraphPad Software Inc., San Diego, CA, USA). The level of statistical significance was set at *p* < 0.05. The figures were also prepared using GraphPad Prism software.

## 3. Results

### 3.1. Establishment of Oxidative Stress Model Induced by DQ in IPEC-J2 Cells

The oxidative stress model of IPEC-J2 cells was established by the induction of DQ. IC50 represents 50% of the inhibitor concentration required for the inhibition of cell viability and enzymes activity [24]. The IPEC-J2 cells’ viability was significantly reduced by 100 μM of DQ and reduced to less than 50% of the control group by DQ of 1250 μM (Figure 1A). Notably, DQ reduced the IPEC-J2 cells’ viability in a dose-dependent manner. The IC50 of DQ for IPEC-J2 cells was approximately 1127.12 μM. Therefore, the concentration of 1150 μM was used as the optimal dose of DQ for the following experiments.

IPEC-J2 cells were pretreated with acidifiers at different concentrations (0, 5 × 10^−5^, 1 × 10^−4^, 5 × 10^−4^, 1 × 10^−3^, 1.5 × 10^−3^, and 2 × 10^−3^) and incubation times (2 h, 4 h, and 6 h), and then their cell viability was further detected. As shown in Figure 1B, both the high concentration of acidifier (i.e., 1 × 10^−3^) and the longer incubation time decreased cell viability (Figure 1B), while pretreatment with low concentrations of acidifier (e.g., 5 × 10^−5^) significantly increased cell viability. Specifically, the cell viability was significantly increased by pretreatment with acidifiers for 4 h. The cells that were pretreated with 5 × 10^−5^ acidifier had the greatest cell viability. Hence, the experimental parameters (5 × 10^−5^ acidifier concentration and 4 h pretreatment time) were used in the subsequent experiments.

### 3.2. Effects of Acidifier on ROS Production and Antioxidant Capacity in DQ-Induced IPEC-J2 Cells

Oxidative stress is caused by an imbalance between the production and clearance of ROS [25]. We first examined the effects of acidifiers on ROS production in DQ-induced IPEC-J2 cells. DQ increased the generation of ROS compared with the CON group (*p* < 0.05), while acidifier markedly inhibited the generation of ROS (*p* < 0.05) (Figure 2A). At the same time, DQ increased the concentration of MDA, while acidifier+DQ (AC + DQ group) decreased the concentration of MDA (Figure 2B). In addition, the activity of CAT was decreased (*p* < 0.05) by 34.39% in the DQ group compared with the AC group (Figure 2C). Moreover, the activities of T-SOD and GSH-Px in the DQ group were lesser (*p* < 0.05) than those of the CON group, while those in the AC + DQ group were greater than those of the DQ group (Figure 2D,E). However, no significant difference in the activity of T-AOC was observed between the DQ and CON groups, but it was increased in the AC group compared with the CON group (*p* < 0.05) (Figure 2F).

### 3.3. Protective Effect of Acidifiers on IPEC-J2 Cells Exposure to Oxidative Stress

Intestinal epithelial cells have barrier functions such as tight junctions, preventing the invasion of certain toxins from penetrating the intestine [26]. Next, we further investigated the protective potential of acidifiers on the barrier function of small intestinal epithelial cells after oxidative stress induced by DQ. DQ challenge decreased (*p* < 0.05) the abundance of tight-junction proteins (Claudin-1) (Figure 3A,B). DQ decreased the mRNA expression of *claudin-1* in the IPEC-J2 cells compared with the CON treatment (*p* < 0.05) (Figure 3C), whereas the acidifiers increased the mRNA expression of *claudin-1* in the DQ-induced cells (*p* < 0.05) (Figure 3C). Similarly, the cellular mRNA expression of *occludin* was also downregulated in the DQ group compared with those in the CON group (*p* < 0.05) (Figure 3D). The cellular expression of *occludin* in the AC + DQ group was greater than that in the DQ group (*p* < 0.05) (Figure 3D). However, no significant difference was found in the mRNA level of *ZO-1* between treatment groups (Figure 3E).

### 3.4. Effects of Acidifiers on the Inflammatory Cytokines in IPEC-J2 Cells

Oxidative stress has been linked to inflammation response and may be a defining feature of inflammation [27]. We next investigated the secretion and gene expression levels of cytokines involved in the inflammatory response. As shown in Figure 4A–C, IPEC-J2 cells stimulated with diquat alone secreted greater levels of pro-inflammatory cytokine TNF-α and IL-8, and a lower level of anti-inflammatory cytokine IL-10 in comparison with control group (*p* < 0.05). However, acidifiers reduced the secretion levels of pro-inflammatory cytokine TNF-α and IL-8 and increased IL-10 secretion level in the DQ-induced IPEC-J2 cells culture supernatant (Figure 4A–C). In addition, DQ increased the mRNA expression levels of *TNF-α* and *IL-8* genes compared with those in the CON group (*p* < 0.05) (Figure 4D,E). However, acidifiers downregulated their expression in the DQ-induced IPEC-J2 cells (*p* < 0.05) (Figure 4D,E). In contrast, a lower expression level of *IL-10* was detected in the DQ group than that in the CON group (*p* < 0.05). In addition, the *IL-10* gene expression level was upregulated in the DQ+AC group compared with that in the DQ group (*p* < 0.05) (Figure 4F).

### 3.5. Acidifiers Alleviated Oxidative Stress-Induced Apoptosis in IPEC-J2 Cells

Oxidative stress triggers apoptosis in different types of cells, including intestinal epithelial cells [28]. We examined the effect of acidifiers on the apoptosis of IPEC-J2 cells induced by DQ. We first examined the IPEC-J2 cells apoptosis by Annexin-V staining and a flow cytometry analysis. The percentages of apoptotic cells in the CON, DQ, AC + DQ, and AC groups were 5.05 ± 1.31%, 12.95 ± 0.64%, 5.67 ± 0.45%, and 6.48 ± 0.40%, respectively (Figure 5A). DQ increased the apoptosis rate of IPEC-J2 cells, while acidifiers dramatically downregulated DQ-induced apoptosis (*p* < 0.05) (Figure 5B). Then, we detected the expression levels of apoptosis-related genes. The gene expression level of *BAX* was increased in the IPEC-J2 cells treated with DQ compared with those untreated cells, but acidifiers reversed this increase (*p* < 0.05) (Figure 5C). In addition, DQ markedly downregulated the gene expression level of *BCL2* compared with the CON group (*p* < 0.05), while acidifiers upregulated the gene expression level of *BCL2* compared with the DQ group (*p* < 0.05) (Figure 5D). In general, these findings suggest that the acidifiers might alleviate apoptosis triggered by DQ in IPEC-J2 cells.

### 3.6. Effect of Acidifiers on the Proliferation of IPEC-J2 Cells Induced by DQ

At the cellular level, oxidative stress also triggers a wide range of responses, including cell proliferation [29]. To further demonstrate whether the acidifiers can rescue the DQ-induced inhibition of proliferation in IPEC-J2 cells, cell proliferation assays were conducted. The percentage of Edu-positive fluorescent cells in the DQ group was less than that in the CON group (*p* < 0.05), while the acidifiers increased the Edu-positive fluorescent cells (*p* < 0.05) and rescued the DQ-induced inhibition. (Figure 6A,B). Since the *CCND1* and *PCNA* genes are related to cell proliferation, we examined the expression levels of these genes. DQ decreased the mRNA levels of *PCNA* and *CCND1* genes (*p* < 0.05), while the acidifiers increased the mRNA levels (*p* < 0.05) and rescued DQ-induced inhibition (Figure 6C,D). The fluorescence intensity of IPEC-J2 cells was reduced after DQ treatment (*p* < 0.05) (Figure 6A), indicating that DQ reduced DNA replication activity. All these results indicate that the acidifiers can protect intestinal cells against DQ-induced oxidative stress by regulating proliferation and apoptosis-related proteins.

### 3.7. Effects of Acidifiers on the Activation of the NF-κB/MAPK/COX-2 Signaling Pathways in DQ-Induced IPEC-J2 Cells

To investigate whether the acidifiers can attenuate DQ-induced oxidative stress mediated by the NF-κB/MAPK/COX-2 signaling pathways, we determined the mRNA and protein expression levels of the COX-2, NF-κB, I-κB-α, I-κB-β, ERK1, and JNK2 in IPEC-J2 cells. As shown in Figure 7, we found that DQ upregulated COX-2, NF-κB, I-κB-β, ERK1, and JNK2 at both the mRNA (Figure 7A,B,G–I) and protein levels (Figure 7D,E,J–L). In contrast, the mRNA (Figure 7A,B,G–I) and protein (Figure 7D,E,J–L) levels of the COX-2, NF-κB, I-κB-β, ERK1, and JNK2 in the AC + DQ group were lower than those in the DQ group (*p* < 0.05). Moreover, DQ decreased the mRNA and protein expression levels of I-κB-α (Figure 7C,F). Importantly, the acidifiers rescued the mRNA and protein levels of I-κB-α in the DQ-induced IPEC-J2 cells (Figure 7C,F).

## 4. Discussion

Acidifiers, as vital feed additives in newly weaned piglets, play a critical role in the improvement of growth performance and the intestinal microbiota in pigs [30]. Our previous study found that supplementing a type of acidifier called *Jinlisuan* in drinking water increased the antioxidant capacity and immunity of weaned pigs [15]. However, its potential protective effects on intestinal epithelial cells and underlying molecular mechanism are not clear, especially after intestinal epithelial cells suffer from oxidative stress and present an inflammatory response [31]. Understanding the protective effects of acidifiers on small intestinal epithelial cells and their potential antioxidant mechanisms is important for their application in pig-feed additives. The IPEC-J2 cell line was isolated from the mid-jejunum of newborn piglets. IPEC-J2 cells have the ability to differentiate and are similar to primary intestinal epithelial cells [32]. DQ is a classic oxidative stress inducer that causes oxidative stress and cell dysfunction by producing reactive oxygen species and reactive nitrogen [33]. In this study, the IPEC-J2 cell line was used as a model cell line to investigate the antioxidative activity of acidifiers, and diquat was used as a stimulant to construct an intestinal epithelial exogenous oxidative stress model. Therefore, we established a DQ-induced oxidative stress model in the IPEC-J2 cells and evaluated the protective effects of the acidifiers on reducing the oxidative stress and inflammatory response in the IPEC-J2 cells.

The acidifier used in this study is a water-soluble liquid acidifier. The acidifier is composed of a mixture of organic acids such as formic acid, acetic acid, propionic acid, lactic acid and their salts in this study. The doses of liquid-type organic acid mixture are generally expressed by weight percentage in water [34]. In this study, we used the culture medium to dilute the liquid acidifier to analyze the optimal dosage in vitro. In the previous animal experiment, the purpose of diluting the acidifier was also achieved by adding liquid acidifier to drinking water [15]. Therefore, the effectiveness of an acidifier is determined by the dilution ratio. Previous studies have shown that acidifiers protected intestinal epithelial cells against damage and improved cell viability and proliferation [35]. Consistent with these results, the present study showed that the final dilution ratio of acidifiers of 1:2 × 10^5^ can improve cell viability in DQ-treated IPEC-J2 cells.

Importantly, we also found that high concentrations of acidifiers inhibited the cell viability of IPEC-J2 cells, while low concentrations of acidifiers increased cell viability. It was reported that low concentrations of organic acid promoted the differentiation of IPEC-J2 cells, while a higher concentration of organic acid impaired cell viability and inhibited cell proliferation in a dose-dependent manner [36]. The acidifiers used in the present study were weak organic acids, including formic acid, acetic acid, propionic acid, and lactic acid [37]. Therefore, we hypothesized that acidifiers improve cell viability at appropriate concentrations, but they can also cause severe damage to cells at excessive concentrations. Our experiment demonstrated that treatment of the acidifiers with appropriate concentrations and time enhanced the viability of IPEC-J2 cells.

ROS are important markers of oxidative stress, which are usually used to evaluate oxidative stress [38]. MDA is a lipid peroxidation biomarker that reflects oxidative damage [39]. To protect cells against ROS-induced oxidative damage, antioxidant systems including CAT, SOD, and GSH-Px are subsequently activated [40]. Importantly, total antioxidant capacity (T-AOC) refers to the total antioxidant level composed of various antioxidant substances and antioxidant enzymes, which help to protect cells and the body from oxidative stress damage caused by reactive oxygen free radicals [41]. Therefore, total antioxidant capacity is of great significance to scientifically evaluate the antioxidant capacity of antioxidant substances. In the present study, DQ increased ROS and MDA levels in IPEC-J2 cells, while the acidifiers decreased the levels of ROS and MDA by reversing the decrease in T-SOD and GSH-Px, suggesting that the acidifiers have potential antioxidant effects under cellular oxidative stress, which is consistent with previous studies [42,43].

Oxidative damage caused by oxidative stress increases the permeability of intestinal epithelial cells [44]. The increase in intestinal permeability impaired the integrity of the intestinal epithelial barrier [45]. Cell permeability is determined by the tight-junction proteins, including claudins, occludin, and ZO-1, which separate the internal environment from the external environment and block harmful substances [46]. A previous study revealed that sodium butyrate decreased the permeability and selectively increased the expression level of tight-junction proteins in IPEC-J2 cells [47]. Indole-3-propionic acid improved the expression of tight-junction proteins (claudin-1, occludin, and ZO-1) and the intestinal epithelial barrier [48]. Our present study is consistent with these findings, which suggest that the acidifiers can maintain epithelial integrity by preventing the increase in cell permeability caused by oxidative stress.

The inflammatory response is a defining feature of oxidative stress caused by ROS [49] because inflammatory factors are involved in biochemical reactions that produce ROS [50]. IL-8, IL-10, and TNF-α are inflammatory response markers [51]. Diquat is used as a model inducer of oxidative stress and induces an inflammatory response, which manifests as a decrease in the mRNA expression of *IL-10* and an increase in *TNF-α* mRNA expression in weaned piglets [52]. In the present study, DQ induced an inflammatory response in IPEC-J2 cells by stimulating the production of proinflammatory cytokines such as TNF-α and IL-8, while inhibiting the production of anti-inflammatory cytokine IL-10. The acidifiers, on the other hand, downregulated the expression of *TNF-α* and *IL-8* genes in the DQ-induced IPEC-J2 cells. Previous research has shown that the supplementation of organic acids reduces the concentration of pro-inflammatory cytokines IL-12 and IL-16, as well as inflammatory biomarker Pentraxin-3 in serum [53], which is consistent with our present study. These results suggest that the acidifiers can attenuate the DQ-induced cellular inflammatory response by decreasing the production of pro-inflammatory cytokines and increasing the production of anti-inflammatory cytokines.

Oxidative stress triggers apoptosis of the intestinal epithelial cells [54]. BCL-2 family members are apoptosis markers, in which BAX promotes apoptosis while BCL-2 inhibits apoptosis [55]. A previous study reported DQ-induced hepatic apoptosis and mitochondrial dysfunction in piglets [56]. In our present study, we found that IPEC-J2 cells treated with diquat increased the mRNA expression of *BAX* and decreased the mRNA expression of *BCL-2*, indicating that DQ induced IPEC-J2 cell apoptosis. IPEC-J2 cells pretreated with the acidifiers significantly increased the mRNA expression of *BCL-2* compared with the DQ-treated group. Similarly, acidifiers markedly upregulated the mRNA level of *BCL-2* compared with untreated cells, while no significant changes were found compared with the acidifiers pre-treatment (AC + DQ) group. These results indicated that pretreatment with acidifiers could reduce the apoptosis of IPEC-J2 cells via upregulating *BCL-2* expression. Thus, the acidifiers rescued the apoptosis of IPEC-J2 cells by regulating the expression of apoptosis-related genes. In addition, oxidative stress is closely related to cell proliferation [57]. In mice, for example, hydrogen treatment preserved intestinal epithelial cell proliferation while reducing oxidative stress damage and the systemic inflammatory response [58]. As expected, our present study revealed that DQ decreased the percentage of Edu-positive fluorescent cells while the acidifiers rescued them. Similarly, the expression levels of genes (*PCNA* and *CCND1*) related to proliferation were decreased in the DQ group compared with those in the CON group, while the acidifiers rescued the mRNA expression levels of the *PCNA* and *CCND1* genes. Taken together, these results suggest that the acidifiers can increase DNA replication activity and are capable of promoting IPEC-J2 cells proliferation.

Organic acid reduces oxidative stress damage and inflammation by modulating the activity of oxidative stress signaling pathways [59]. For example, oxidative stress activates the inflammatory response via redox regulation of the inhibitor(I)-κB/nuclear factor (NF)-κB signaling pathway [60]. In the present study, DQ increased the mRNA and protein expression levels of NF-κB and IκB in the IPEC-J2 cells, whereas the acidifiers decreased them. This indicated that DQ activated the NF-κB signaling pathway in the IPEC-J2 cells, while acidifiers led to the inhibition of the NF-B pathway, as well as a decrease in NF-κB-mediated transcription. On the other hand, inhibition of the NF-κB pathway leads to the reduction in proinflammatory cytokines [61], which may be the reason for the changes in cytokine levels in the present study. It has been reported that the activation of the mitogen-activated protein kinase (MAPK) pathway is highly correlated with the NF-κB pathway [62]. The MAPK signal pathway (ERK, JNK, p38) is involved in the regulation of cell growth, environmental adaptation to stress, and inflammatory reactions [63], which is also closely associated with oxidative stress [64]. In the present study, the acidifiers attenuated DQ-induced oxidative stress by downregulating the expression levels of proteins (ERK1/2 and JNK2) related to the MAPK signaling pathway. In addition, the COX-2 signaling pathway is considered to be the most important way to regulate oxidative stress. The activation of this signaling pathway leads to the release of free radicals and the transcription repression of genes encoding antioxidant enzymes [65]. In the present study, the acidifiers downregulated the expression of CoX-2 in the DQ-induced IPEC-J2 cells. It is reported that the inhibition of COX-2 expression reduced the production of ROS and prevented DNA damage [66], which implies that the decrease in COX-2 may be one of the reasons for the downregulation of ROS in the AC + DQ group. Taken together, these results suggest that the acidifiers can attenuate the DQ-induced oxidative stress and inflammatory responses by regulating the NF-κB/MAPK/COX-2 signaling pathways.

## 5. Conclusions

In the present study, DQ induced intestinal epithelial cells oxidative stress and inflammation and increased the apoptosis and permeability of the intestinal epithelial cells. However, the acidifiers not only reduced intracellular ROS, but also attenuated the DQ-induced inflammation and modulated the activities of antioxidant enzymes by regulating the NF-κB/MAPK/COX-2 signaling pathways, indicating the prominent antioxidative capacity of the acidifiers.

## Figures and Tables

**Figure 1 antioxidants-11-02002-f001:**
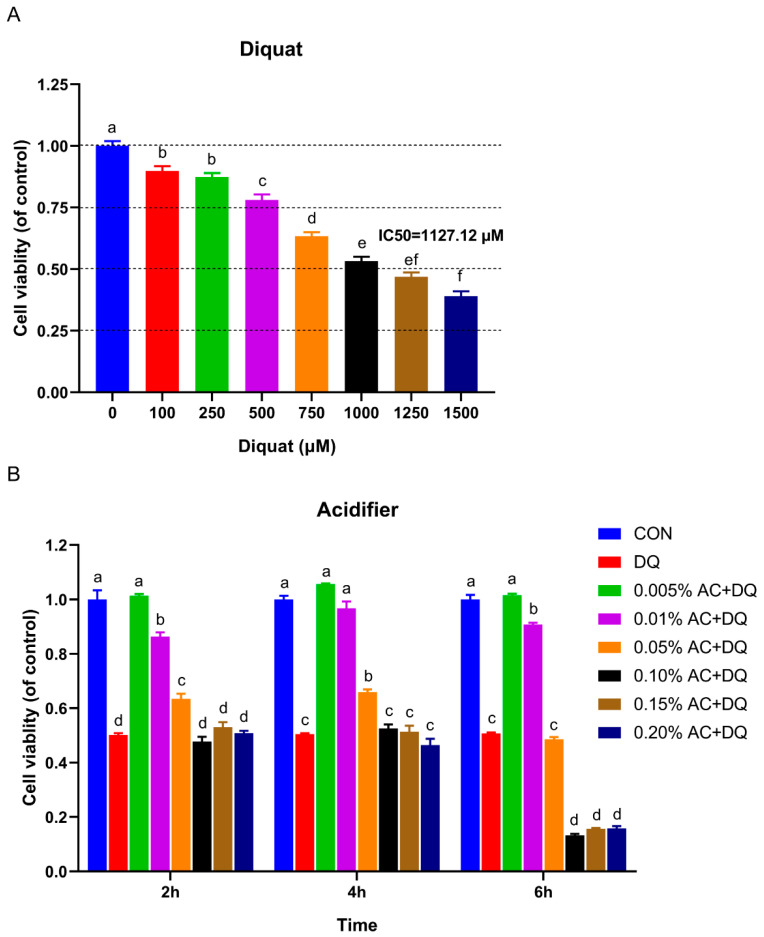
The establishment of an oxidative stress model induced by diquat. (**A**) Effects of different concentrations of diquat on the viability of IPEC-J2 cells for 6 h. The IC50 was calculated using GraphPad Prism 8 software. Cell cytotoxicity was evaluated utilizing an MTT cell assay kit. (**B**) Effects of acidifier pretreatment with different concentrations on the viability of IPEC-J2 cells treated with optimal dose of diquat at different times. IPEC-J2 cells were incubated with or without acidifiers at the different concentrations for 2 h, 4 h, or 6 h. Then, the culture medium was replaced with fresh medium containing 1150 μM of diquat. After incubation for 6 h, cell viability was measured by the CCK-8 assay kit. CON, IPEC-J2 cells without being treated; DQ, cells were only treated by 1150 μM diquat. AC + DQ at 0.005%, 0.01% AC + DQ, 0.05% AC + DQ, 0.1% AC + DQ, 0.15% AC + DQ, 0.20% AC + DQ, cells were pretreated by 0.005%, 0.01%, 0.05%, 0.1%, 0.15%, and 0.20% concentrations acidifiers for 2, 4, 6 h, respectively, and were then treated by 1150 µM of diquat for 6 h. Data were presented as mean ± SEM (*n* = 8). Statistical significance was determined by one-way ANOVA with Tukey’s post hoc test. Different superscript letters denote a statistically significant difference between groups (*p* < 0.05), while same letters denote no significant difference.

**Figure 2 antioxidants-11-02002-f002:**
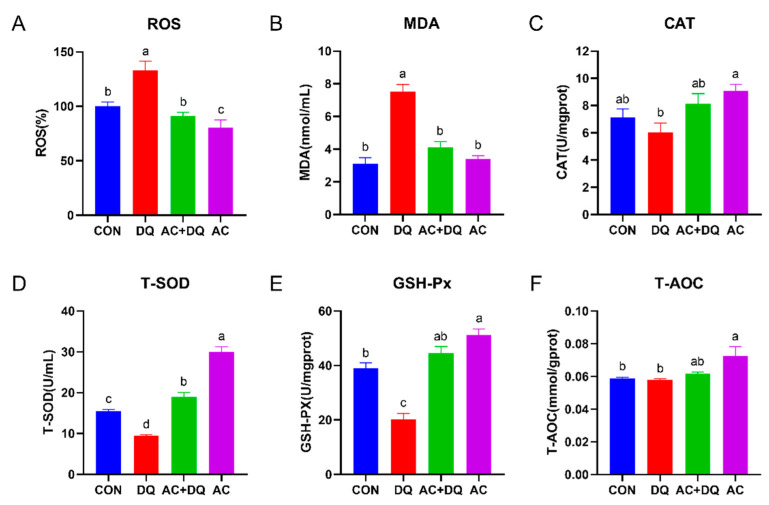
Acidifiers alleviated oxidative stress induced by diquat in IPEC-J2 cells and improved antioxidant capacity. (**A**) Intracellular contents of oxygen reactive species (ROS). (**B**) Cellular MDA levels. (**C**) Cellular CAT activity. (**D**) Cellular T-SOD activity. (**E**) Cellular GSH-Px activity. (**F**) Cellular T-AOC levels. CON, cells without being treated; DQ, cells were only treated by 1150 µM of diquat for 6 h; AC + DQ, cells were pretreated by 0.005% acidifier for 4 h and were then treated by 1150 µM of diquat for 6 h. AC, cells were only treated by 1150 µM of acidifier for 4 h. All values were represented as means ± SEM (*n* = 6). Different superscript letters denote a statistically significant difference between groups (*p* < 0.05), while same letters denote no significant difference.

**Figure 3 antioxidants-11-02002-f003:**
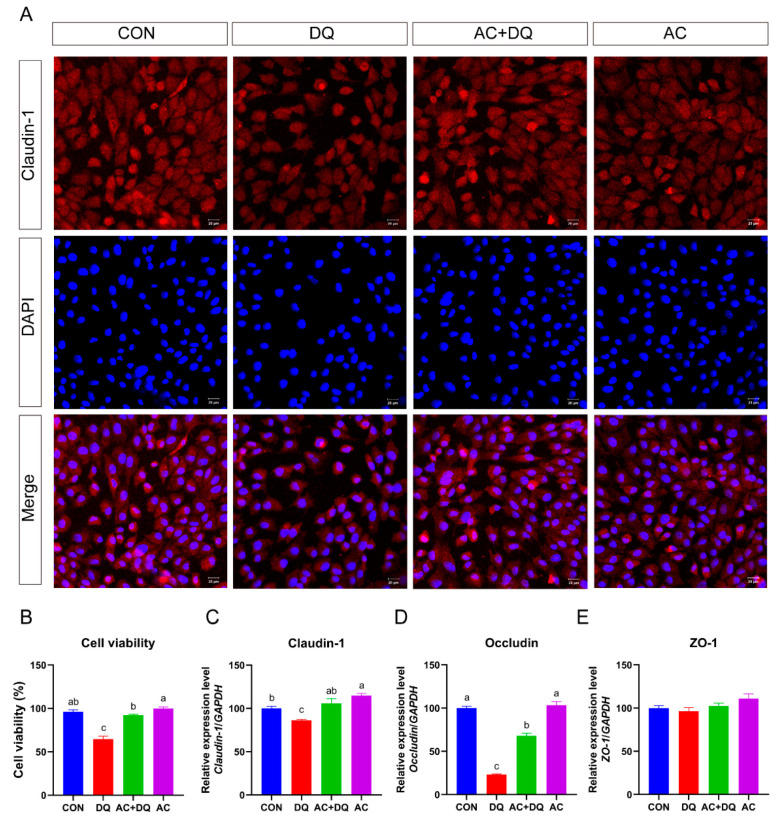
Effects of acidifiers on the expressions of tight-junction-related genes in diquat-challenged IPEC-J2 cells. (**A**) Localization of the tight-junction protein claudin-1 by immunofluorescence staining. IPEC-J2 cells were plated in 12-well plates at a density of 1 × 10^5^ cells per well and pretreated by 0.005% acidifiers for 4 h, and the cells were then challenged by diquat (1150 µM) for 6 h. Immunofluorescence staining of IPEC-J2 cells with claudin-1 (red) and DAPI (blue). Magnification 20×, and scale bars representing 20 μm. (**B**) Assays of cell viability. (**C**) The mRNA expression level of claudin-1 (*CLDN1*). (**D**) The mRNA expression level of occludin (*OCLN*). (**E**) The mRNA expression level of ZO-1. CON, cells without being treated; DQ, cells were only treated by 1150 µM of diquat for 6 h; AC + DQ, cells were pretreated by 0.005% acidifier for 4 h and were then treated by 1150 µM of diquat for 6 h. AC, cells were only treated by 1150 µM of acidifier for 6 h. All values were represented as means ± SEM (*n* = 6). Different superscript letters denote a statistically significant difference between groups (*p* < 0.05), while same letters denote no significant difference.

**Figure 4 antioxidants-11-02002-f004:**
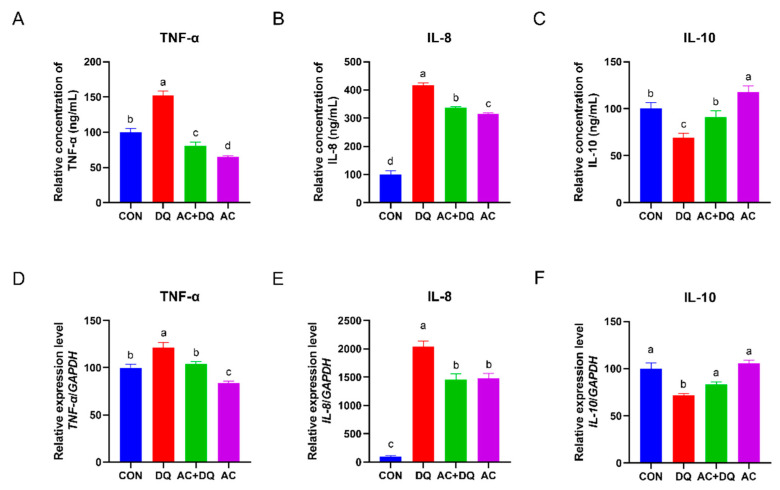
Effects of acidifiers on the secretion and gene expression level of inflammatory cytokines in diquat-challenged IPEC-J2 cells. (**A**) Concentration of TNF-α in the IPEC-J2 cells culture supernatant. (**B**) Concentration of IL-8 in the IPEC-J2 cells culture supernatant. (**C**) Concentration of IL-10 in the IPEC-J2 cells culture supernatant. (**D**) The gene expression level of *TNF-α*. (**E**) The gene expression level of *IL-8*. (**F**) The gene expression level of *IL-10*. CON, cells without being treated; DQ, cells were only treated by 1150 µM of diquat for 6 h; AC + DQ, cells were pretreated by 0.005% acidifier for 4 h and were then treated by 1150 µM of diquat for 6 h. AC, cells were only treated by 1150 µM of acidifier for 6 h. All values were represented as means ± SEM (*n* = 6). Different superscript letters denote a statistically significant difference between groups (*p* < 0.05), while same letters denote no significant difference.

**Figure 5 antioxidants-11-02002-f005:**
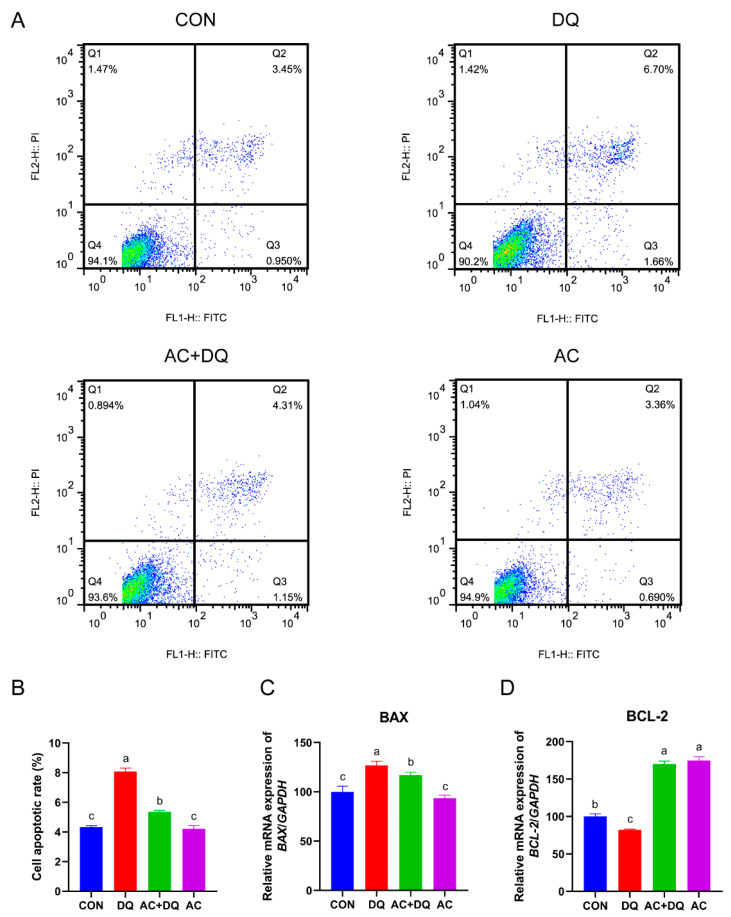
Effect of acidifiers on the apoptosis of IPEC-J2 cells induced by diquat. (**A**) Annexin V-FITC/PI apoptosis assay. Apoptotic cell rates were detected with a FITC annexin V-FITC/PI apoptosis kit, and then analyzed by flow cytometry. (**B**) The cell apoptosis of IPEC-J2 cells. (**C**) The mRNA expression level of *BAX* was determined using a qRT-PCR. (**D**) The mRNA expression level of *BCL-2* was determined using a qRT-PCR. CON, cells without being treated; DQ, cells were only treated by 1150 µM of diquat for 6 h; AC + DQ, cells were pretreated by 0.005% acidifier for 4 h and were then treated by 1150 µM of diquat for 6 h. AC, cells were only treated by 1150 µM of acidifier for 6 h. All values were represented as means ± SEM (*n* = 6). Different superscript letters denote a statistically significant difference between groups (*p* < 0.05), while same letters denote no significant difference.

**Figure 6 antioxidants-11-02002-f006:**
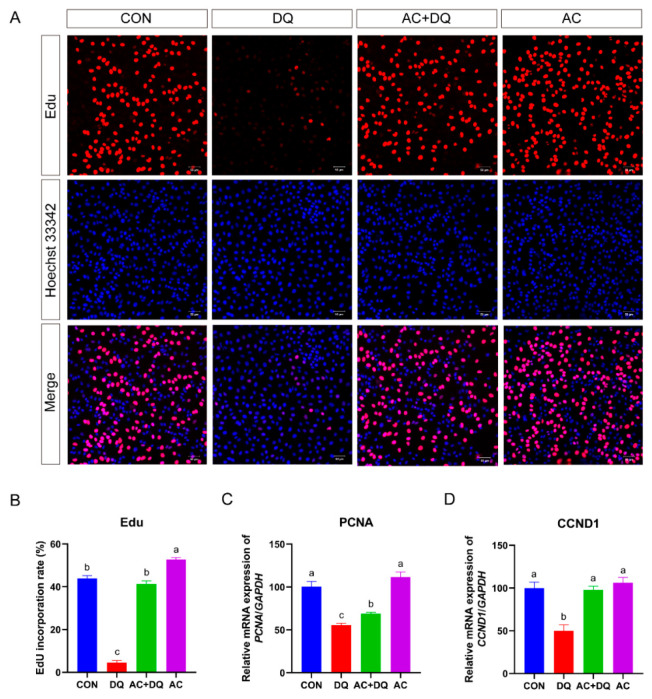
The effect of acidifiers on the proliferation of IPEC-J2 cells induced by diquat. (**A**) Edu staining of IPEC-J2 cells (magnification 10×, scale bar = 50 μm). (**B**) Statistical results of the proportion of EdU-positive cells. (**C**) Relative mRNA level of *PCNA* gene. (**D**) Relative mRNA level of *CCND1* gene. CON, cells without being treated; DQ, cells were only treated by 1150 µM of diquat for 6 h; AC + DQ, cells were pretreated by 0.005% acidifier for 4 h and were then treated by 1150 µM of diquat for 6 h. AC, cells were only treated by 1150 µM of acidifier for 6 h. All values were represented as means ± SEM (*n* = 6). Different superscript letters denote a statistically significant difference between groups (*p* < 0.05), while same letters denote no significant difference.

**Figure 7 antioxidants-11-02002-f007:**
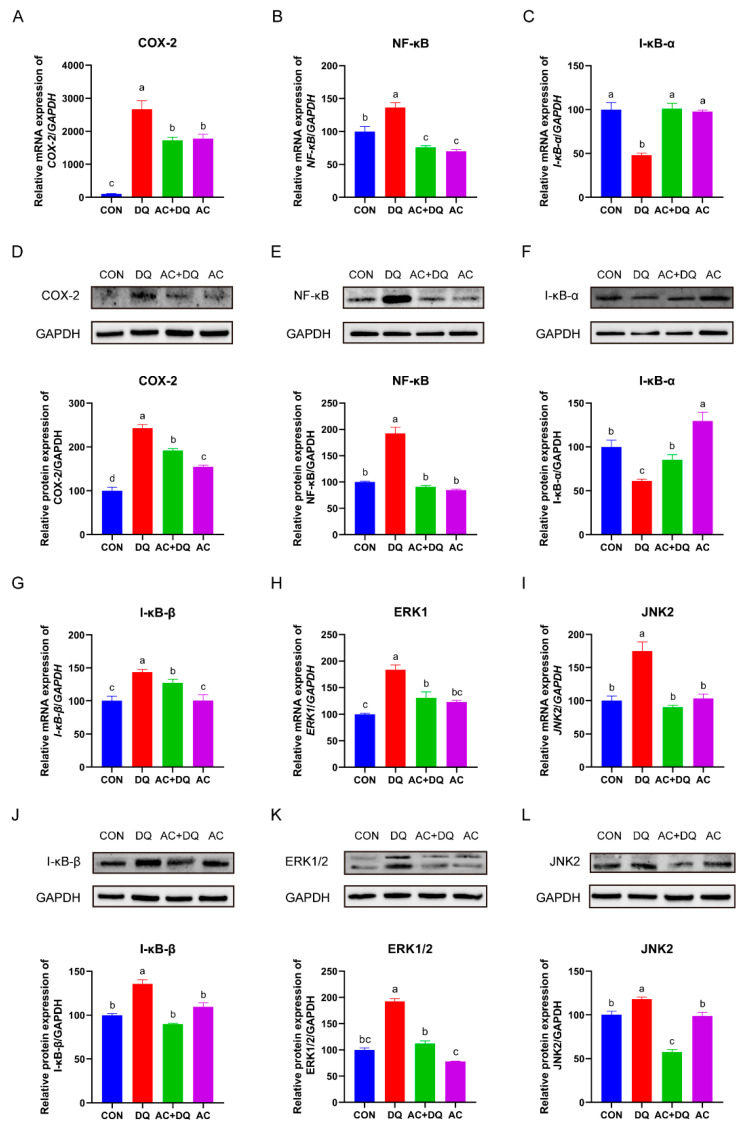
Effects of acidifiers on the NF-κB/MAPK/COX-2 signaling pathways in IPEC-J2 cells. The mRNA levels of *COX-2* (**A**), *NF-κB* (**B**), and *I-κB-α* (**C**). The protein expression level of COX-2 (**D**), NF-κB (**E**), and I-κB-α (**F**). The mRNA levels of *I-κB-β* (**G**), *ERK1* (**H**), and *JNK2* (**I**). The protein expression level of I-κB-β (**J**), ERK1/2 (**K**), and JNK2 (**L**). CON, cells without being treated; DQ, cells were only treated by 1150 µM of diquat for 6 h; AC + DQ, cells were pretreated by 0.005% acidifier for 4 h and were then treated by 1150 µM of diquat for 6 h. AC, cells were only treated by 1150 µM of acidifier for 6 h. All values were represented as means ± SEM (*n* = 3). Different superscript letters denote a statistically significant difference between groups (*p* < 0.05), while same letters denote no significant difference.

## Data Availability

All data are presented in the article and Appendix A.

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
