# Peer review of "Acidifiers Attenuate Diquat-Induced Oxidative Stress and Inflammatory Responses by Regulating NF-κB/MAPK/COX-2 Pathways in IPEC-J2 Cells"

_antioxidants, 2022, doi:10.3390/antiox11102002_

Round 1

Reviewer 1 Report

you have used a certain composition of Acidifiers, could the results obtained be dependent on the composition?

it is possible to motivate why cell viability is greater at low concentrations of acidifiers?

Author Response

Dear Reviewer,

Thank you very much for your constructive comments! We appreciate the editor and the reviewers, who gave us an opportunity to revise our manuscript (antioxidants-1916056).

In the revised manuscript, we have meticulously addressed all the comments raised by the reviewers and associate editor. Our response to the comments was listed point-by-point. All the significant changes in the revised manuscript have been highlighted in red.

Comment 1: you have used a certain composition of Acidifiers, could the results obtained be dependent on the composition?.

Response 1: Thank you for your comments! Yes, it is. Our previous animal testing has shown that this certain composition of acidifiers improved the growth performance of weaned pigs, and improved the antioxidant capacity and immunity in vivo (Xu, et al., Antioxidants, 2022). This study also used the same composition of acidifiers to verify their effectiveness at the cellular level. The results showed that these acidifiers also improved the antioxidant capacity and reduced the inflammatory response in vitro.

Comment 2: it is possible to motivate why cell viability is greater at low concentrations of acidifiers?

Response 2: Thank you for your invaluable comments. The acidifiers used were weakly acidic in this study. Even so, the acidifiers with high concentrations directly reduced cell vitality. When the acidifiers were diluted to low concentrations, the acidifiers improved the cell vitality of small intestinal epithelial cells. In this regard, we have added relevant content to the discussion.

Reviewer 2 Report

The manuscript is written well. The use of English language is very good. The sections could be followed clearly. However, I have the following concerns:

The effects of AC and AC+DQ on BCL-2 mRNA is strange (Fig 5D). It appears a unique mechanism is involved. This experiment should be repeated and explained in the discussion section.

Author Response

Dear Reviewer,

Thank you very much for your constructive comments! We appreciate that the editor and reviewers gave us an opportunity to revise our manuscript (antioxidants-1916056).

In the revised manuscript, we have meticulously addressed all the comments raised by the reviewers and associate editor. Our response to the comments was listed point-by-point. All the significant changes in the revised manuscript have been highlighted in red.

Comment: The manuscript is written well. The use of English language is very good. The sections could be followed clearly. However, I have the following concerns: The effects of AC and AC+DQ on BCL-2 mRNA is strange (Fig 5D). It appears a unique mechanism is involved. This experiment should be repeated and explained in the discussion section.

Response: Thank you for your comments and constructive suggestions! The experiment was repeated and the same result was forthcoming. BCL2 is an antiapoptotic factor that is capable of promoting cell survival. In this study, the mRNA expression of BCL2 was significantly decreased in cells treated with diquat (DQ group). After pretreatment with the acidifiers for 4 h (AC+DQ group), there were significant increases in the expression level of BCL2 compared with the diquat-treated group. Acidifiers (AC group) markedly upregulated the mRNA level of BCL2 compared with untreated cells (CON group), while that in the AC treatment group indicated no significant changes compared with the acidifiers pre-treatment (AC+DQ) group. These results suggest the acidifiers could alleviate the apoptosis of IPEC-J2 cells induced by diquat. The corresponding content has been added in the section of discussion.

Reviewer 3 Report

The manuscript “Acidifiers Attenuate Diquat-induced Oxidative Stress and Inflammatory Responses by Regulating NF-κB/MAPK/COX-2 Pathways in IPEC-J2 Cells” by Qinglei Xu et al. evaluates the protective effects and potential mechanisms of acidifiers on intestinal epithelial cells exposure to oxidative stress.

The goal of the study is to protect piglets at weaning from inflammatory stressors. It is not at all clear that one cell line (IPEC-J2) suffices to model this scenario. The MTT assay is very basic. Diquat is used as a model inducer of oxidative stress, but it is not justified as a suitable model for the inflammation under weaning (diquat induces cytokine secretion, but are these the cytokines produced at weaning?). The acidifier concentrations are not given, only their dilutions. Overall, the scope of the experiments and the details of their description are rather limited. The value of the work for the stated goal is uncertain. More experiments should be done, including additional model systems (in the least, confirmation in additional cell lines) to reach a publication-quality report.

Minor points:

The manuscript contains several typographical and grammatical errors.

It would be helpful to have consistency in the display of significance. Figures 2 and 3, in particular, is difficult to understand. Brackets are more commonly used for similar displays.

Author Response

Dear Reviewer,

Thank you very much for your constructive comments! We appreciate that the editor and reviewers gave us an opportunity to revise our manuscript (antioxidants-1916056).

In the revised manuscript, we have meticulously addressed all the comments raised by the reviewers and associate editor. Our response to the comments was listed point-by-point. All the significant changes in the revised manuscript have been highlighted in red.

Comment 1: The goal of the study is to protect piglets at weaning from inflammatory stressors. It is not at all clear that one cell line (IPEC-J2) suffices to model this scenario. The MTT assay is very basic. Diquat is used as a model inducer of oxidative stress, but it is not justified as a suitable model for the inflammation under weaning (diquat induces cytokine secretion, but are these the cytokines produced at weaning?).

Response 1: Thank you for your excellent comments and constructive suggestions! The purpose of the study was to explore the effect of an acidifier on the antioxidative and inflammatory responses of intestinal porcine epithelial cells (IPEC-J2). Our previous animal testing had shown that acidifiers improved the antioxidant capacity of weaned pigs, immunity and reduced inflammation in vivo (Xu, et al., Antioxidants, 2022). This study aimed to verify the effectiveness of the acidifiers at the cellular level. The results showed that the acidifier also improved the antioxidant capacity and reduced the inflammatory response in vitro. IPEC-J2 cells isolated from the mid jejunum of neonatal piglets may serve as a better resource to explore the effects of acidifiers in vitro, as they have been widely utilized for studying intestinal functions associated with antioxidant system, immune response, and barrier function. In this study, the IPEC-J2 cell line was used as a model cell line to investigate the antioxidative activity of acidifiers, and diquat was used as a stimulant to induce the generation of ROS and inflammatory factors to construct an intestinal epithelial exogenous oxidative stress model, not a weaning stress model of pigs.

Comment 2: The acidifier concentrations are not given, only their dilutions.

Response 2: Thank you for your valuable comments. The acidifier used is a liquid acidifier in this study. The acidifier is composed of organic acids such as formic acid, acetic acid, propionic acid, lactic acid and their salts in this study. The concentration of organic acid is generally expressed by weight percentage in water. In this study, we used the culture medium to dilute the liquid acidifier to analyze the optimal dosage in vitro. In the previous animal experiment, the purpose of diluting the acidifier was achieved by adding liquid acidifier to drinking water. Therefore, the effectiveness of an acidifier is determined by the dilution ratio.

Comment 3: Overall, the scope of the experiments and the details of their description are rather limited. The value of the work for the stated goal is uncertain. More experiments should be done, including additional model systems (in the least, confirmation in additional cell lines) to reach a publication-quality report.

Response 3: Thanks for your comments and constructive suggestions! Our previous study has found that acidifiers could improve the growth performance, antioxidant capacity, and immunity in weaned pigs. This research aimed to verify whether acidifiers can alleviate oxidative stress in IPEC-J2 cells as well as explore its mechanisms in vitro. This will provide a new theoretical support for adding acidifiers as feed additives to piglet diets to improve antioxidant capacity and immunity. We will do the relevant job in the future according to your suggestion.

Comment 4: Minor points: The manuscript contains several typographical and grammatical errors.

Response 4: Thank you for your precise observation! We have carefully reviewed the full text and corrected the typographical and grammatical errors

Comment 5: Minor points: It would be helpful to have consistency in the display of significance. Figures 2 and 3, in particular, is difficult to understand. Brackets are more commonly used for similar displays.

Response 5: Thank you for your comments and constructive suggestions! We have introduced a uniform system for citation p values in the text, use p < 0.05 to indicate a statistically significant difference. Different superscript letters denote a statistically significant difference between groups (p < 0.05), while the same letters denote no significant difference.

Round 2

Reviewer 3 Report

The original review recommended major revision. The authors have chosen not to undertake more than minimal changes. 

Author Response

Dear Reviewer,

Thank you very much for your constructive comments! We appreciate that the Editor and Reviewers gave us an opportunity to revise our manuscript (antioxidants-1916056).

In the revised manuscript, we have meticulously addressed all the comments raised by the reviewers and associate editor. Our response to the comments was listed point-by-point. All the significant changes in the revised manuscript have been highlighted in red.

Comment 1: The goal of the study is to protect piglets at weaning from inflammatory stressors. It is not at all clear that one cell line (IPEC-J2) suffices to model this scenario. Diquat is used as a model inducer of oxidative stress, but it is not justified as a suitable model for the inflammation under weaning (diquat induces cytokine secretion, but are these the cytokines produced at weaning?).

Response 1: Thank you for your excellent comments and constructive suggestions! IPEC-J2 cells isolated from the mid jejunum of neonatal piglets may serve as a better resource to explore the effects of acidifiers in vitro, as they have been widely utilized for studying intestinal functions associated with antioxidant system, immune response, and barrier function. In this study, the IPEC-J2 cell line was used as a model cell line explore the effect of an acidifier on the antioxidative and inflammatory responses of intestinal porcine epithelial cells (IPEC-J2). Our previous animal testing had shown that acidifiers improved the antioxidant capacity of weaned pigs, immunity and reduced inflammation in vivo (Xu, et al., Antioxidants, 2022). This study aimed to verify the effectiveness of the acidifiers at the cellular level. The results showed that the acidifier also improved the antioxidant capacity and reduced the inflammatory response in vitro.

Diquat-challenged inducing intestinal inflammation and oxidative stress in weaned piglets. Diquat induces an inflammatory response, which manifests as decrease the mRNA expression of IL-10 and increase of TNF-α mRNA expression in weaned piglets. So, diquat was used as a stimulant to induce the generation of ROS and inflammatory factors to construct an intestinal epithelial exogenous oxidative stress model, not a weaning stress model of pigs. The corresponding content has been added in the section of discussion and identified in red font.

Comment 2: The acidifier concentrations are not given, only their dilutions.

Response 2: Thank you for your valuable comments. The acidifier used is a liquid acidifier in this study. The acidifier is com-posed of the mixture of organic acids such as formic acid, acetic acid, propionic acid, lactic acid and their salts in this study. The doses of liquid-type organic acid mixture are generally expressed by weight percentage in water. In this study, we used the culture medium to dilute the liquid acidifier to analyze the optimal dosage in vitro. In the previous animal experiment, the purpose of diluting the acidifier was also achieved by adding liquid acidifier to drinking water. Therefore, the effectiveness of an acidifier is determined by the dilution ratio. The corresponding content has been added in the section of discussion and identified in red font.

Comment 3: Overall, the scope of the experiments and the details of their description are rather limited. The value of the work for the stated goal is uncertain. More experiments should be done, including additional model systems (in the least, confirmation in additional cell lines) to reach a publication-quality report.

Response 3: Thanks for your comments and constructive suggestions! Our previous study has found that acidifiers could improve the growth performance, antioxidant capacity, and immunity in weaned pigs. This research aimed to verify whether acidifiers can alleviate oxidative stress in IPEC-J2 cells as well as explore its mechanisms in vitro. This will provide a new theoretical support for adding acidifiers as feed additives to piglet diets to improve antioxidant capacity and immunity. We will do the relevant job in the future according to your suggestion.

Comment 4: Minor points: The manuscript contains several typographical and grammatical errors.

Response 4: Thank you for your precise observation! We have carefully reviewed the full text and corrected the typographical and grammatical errors.

Comment 5: Minor points: It would be helpful to have consistency in the display of significance. Figures 2 and 3, in particular, is difficult to understand. Brackets are more commonly used for similar displays.

Response 5: Thank you for your comments and constructive suggestions! We have introduced a uniform system for citation p values in the text, use p < 0.05 to indicate a statistically significant difference. Different superscript letters denote a statistically significant difference between groups (p < 0.05), while the same letters denote no significant difference. The corresponding content has been added to the original manuscript and identified in red font.